# Understanding of Ovarian Cancer Cell-Derived Exosome Tropism for Future Therapeutic Applications

**DOI:** 10.3390/ijms24098166

**Published:** 2023-05-03

**Authors:** Xiaoyu Ren, Changsun Kang, Lucila Garcia-Contreras, Dongin Kim

**Affiliations:** 1Department of Pharmaceutical Sciences, College of Pharmacy, University of Oklahoma Health Sciences Center, Oklahoma City, OK 73117, USA; xiaoyu-ren@ouhsc.edu (X.R.); chang-kang@ouhsc.edu (C.K.); lucila-garcia-contreras@ouhsc.edu (L.G.-C.); 2Stephenson Cancer Center, University of Oklahoma Health Sciences Center, Oklahoma City, OK 73104, USA

**Keywords:** exosome, quantification, tropism, ovarian cancer

## Abstract

Exosomes, a subtype of extracellular vesicles, ranging from 50 to 200 nm in diameter, and mediate cell-to-cell communication in normal biological and pathological processes. Exosomes derived from tumors have multiple functions in cancer progression, resistance, and metastasis through cancer exosome-derived tropism. However, there is no quantitative information on cancer exosome-derived tropism. Such data would be highly beneficial to guide cancer therapy by inhibiting exosome release and/or uptake. Using two fluorescent protein (mKate2) transfected ovarian cancer cell lines (OVCA4 and OVCA8), cancer exosome tropism was quantified by measuring the released exosome from ovarian cancer cells and determining the uptake of exosomes into parental ovarian cancer cells, 3D spheroids, and tumors in tumor-bearing mice. The OVCA4 cells release 50 to 200 exosomes per cell, and the OVCA8 cells do 300 to 560 per cell. The uptake of exosomes by parental ovarian cancer cells is many-fold higher than by non-parental cells. In tumor-bearing mice, most exosomes are homing to the parent cancer rather than other tissues. We successfully quantified exosome release and uptake by the parent cancer cells, further proving the tropism of cancer cell-derived exosomes. The results implied that cancer exosome tropism could provide useful information for future cancer therapeutic applications.

## 1. Introduction

All cells, whether healthy or abnormal, release extracellular vesicles (EVs) that mediate cell-to-cell communication in biological processes, whether normal or pathological [1]. There are several subtypes of EV, classified based on their size and cellular origin; one such subtype is exosomes, which range from 50–200 nm in diameter [2,3,4]. The biogenesis of exosomes starts at the inward plasma membrane, where intraluminal vesicles (ILVs) form within the early endosomes [4,5]. Within the endosomes, various proteins and macromolecules are deposited into ILVs, and the early endosomes mature sequentially into late endosomes and then multivesicular bodies (MVBs). Finally, the MVBs are either degraded by lysosomes or fused with the plasma membrane and are released as EVs, including microvesicles and exosomes.

Exosomes derived from tumors have diverse functions in the tumor progression [6,7], chemotherapy resistance [8], and metastasis [9,10,11]. Tumor-derived exosomes contain proteins, lipids, microRNAs, and mRNAs that can be transferred to recipient cells by fusion of the exosomes with the target cell membranes, indicating that tumor-derived exosomes are critical mediators of tumorigenesis [12].

Interestingly, recent studies show compelling evidence of the homing property of tumor-derived exosomes, a phenomenon called “tumor-derived exosome tropism” [13,14,15]. Moreover, compared with non-tumor organs, markedly greater amounts of tumor-derived exosomes target their parental cancer cells due to such tropism [13,14,15,16]. This tropism profoundly influences tumor proliferation, drug resistance, and tumor metastasis.

Exosomes released from tumor cells contain various oncogenic cargos, which can be delivered to neighboring tumor cells through internalization of the exosomes, thus restoring missing functions [17]. For instance, Al-Nedawi et al. reported that mutant epidermal growth factor receptor (EGFRvIII), residing on the membrane of exosomes derived from glioma cells, can be delivered to neighboring glioma cells that lack the mutant form. This augments both expressions of anti-apoptotic genes and tumor cell growth [18]. Similar findings have been reported in colon cancer cells with mutant KRAS proteins [19]. Exosomes derived from colon cancer cells with mutant KRAS alleles delivered their cargo to adjacent colon cancer cells, inducing the expression of mutant KRAS protein and causing tumor cell growth and tumorigenicity. Another study in hepatocellular carcinoma (HCC) showed that the transfer of miRNA to the parental cells via exosomes contributed to multifocal tumor growth by decreasing the expression of transforming growth factor beta-activated kinase-1 (TAK1) [20,21], establishing it as a potentially critical target candidate for regulation via exosomal miRNA [22].

Tumor-derived exosomes also mediate tumor resistance by transferring resistance to sensitive tumor cells [23,24,25,26]. One major multidrug-resistance (MDR) mechanism is the increased expression of P-glycoprotein (P-gp), also known as MDR protein 1 (MDR-1), which functions as a drug efflux agent on the cancer cell membrane and exports substances, including therapeutic drugs, from the intracellular milieu to the extracellular matrix [27,28]. Direct transfer of P-gp or induction of its gene expression in sensitive cancer cells via the internalization of MDR cancer cell exosomes is a major mechanism of tumor resistance. Lv et al. found that exosomes from MDR MCF-7 breast cancer cells transferred drug resistance to sensitive MCF-7 cells [29]. Similarly, Corcoran et al. demonstrated the transfer of P-gp proteins from MDR prostate cancer cells to sensitive cancer cells via exosomes led to acquired docetaxel resistance in the sensitive prostate cancer cells [30]. Another interesting study showed that human MDR osteosarcoma (MG-63) cell-derived exosomes could transfer P-gp mRNA to sensitive osteosarcoma (MG-63) cells, thus conferring MDR on them [31].

Tumor-derived exosomes also promote the metastatic tumor-cell phenotype by transferring various genes and proteins to non-invasive tumor cells [31,32]. Upon internalization of tumor-derived exosomes, non-invasive or non-metastatic tumor cells are inclined to have more epithelial properties, and modulation of such tumors to become metastatic is regulated by the degree of epithelial-to-mesenchymal transition (EMT). Thus, the tropism of exosomes released from active metastatic tumors towards dormant tumor cells increases EMT and promotes tumor metastasis. Various cargos of metastatic tumor-derived exosomes likely promote tumor metastasis. For example, microRNAs (miR-200c/miR0141) promote the metastasis of breast cancer tumors by arousing dormant breast cancer cells [33]. Similarly, Fu et al. have shown that exosomes from HCC transfer SMAD family member 3 (SMAD3) protein to recipient HCC cells and thus enhance SMAD3-related signaling pathways to promote cell adhesion in the lung metastasis [34].

Although it is well established that tumor-derived exosomes promote tumor survival mechanisms through their cancer cell tropism, there is no precise quantitative data regarding the tumor exosome tropism process, i.e., the number of exosomes released by a single tumor cell or the number of exosomes internalized by a single tumor cell. Moreover, such quantitative information is critical for understanding tumor survival mechanisms and could also guide cancer therapy when delivering drugs in an exosome-based drug delivery system. Since the absolute amounts of cancer cell-derived exosomes homing to their parental cells remain obscure, this study reveals important proof-of-concept data on cancer cell-derived exosome tropism. Herein, we establish how many exosomes are released from a single tumor cell and how many released exosomes home to their parental cells due to tropism. Our data show that markedly greater numbers of ovarian cancer cell-derived exosomes are homing to the parental cells than to non-tumor ovarian epithelial cells. Our results also indicate that the quantitative information of tumor-derived exosomes can likely be useful for cancer therapy, such as the co-treatment of exosome inhibitors with cancer drugs.

## 2. Results

### 2.1. Confirmation of mKate2 Protein Transfection in Both OVCA4 and OVCA8 Cells

In order to quantify the number of exosomes released from cells and taken up by cells, we needed to construct fluorescently-labeled cells to produce fluorescently-labeled exosomes. Thus, in this study, we transfected two ovarian cancer cell lines, OVCA4 and OVCA8; OVCA4 is a high-grade serous ovarian carcinoma cell line, and OVCA8 cell line is a non-serous tumor cell line, both of which were fused with mKate2 protein to create fluorescent OVCA4 and OVCA8 cells [35]. After using a cationic lipid transfection reagent (Lipofectamine^TM^ 3000 reagent), the mKate2 protein carried by a mammalian expression vector (FP181) was transfected into the cell lines by endocytosis [36,37,38]. Confocal images of transfected OVCA4 (Figure 1A) and OVCA8 (Figure 1C) showed nuclei (stained blue with DAPI) and red fluorescent mKate2 protein, indicating that mKate2 proteins were successfully transfected in both cell lines. A simple linear regression of fluorescence of mKate2 protein on both OVCA4-mKate2 (Figure 1B, *n* = 3–4) and OVCA8-mKate2 (Figure 1D, *n* = 5) with known numbers of cells was performed and showed a linear relation of mKate2 protein fluorescence with numbers of OVCA4-mKate2 and OVCA8-mKate2 ovarian cancer cells.

We next used 3D tumor spheroid models to quantify exosomal tropism. Compared to attached cells, spheroids are more likely to mimic tumor cells in vivo, including with respect to cell morphology and polarity, the interaction between cells, and the interaction between cells and the extracellular matrix [39]. Confocal images of spheroids of OVCA4-mKate2 (Figure 1E) and OVCA8-mKate2 (Figure 1F) cells show nuclei stained blue with DAPI and fluorescence of mKate2 protein in red. Confocal images were obtained from at least three independent preparations. Western blotting of cell samples detected mKate2 in both OVCA4-mKate2 and OVCA8-mKate2 cells, migrating at ~27 kDa (Figure 1G). Western blots were performed with four independent preparations. Observation of fluorescence of mKate2 protein by confocal microscopy and detecting mKate2 protein by western blot indicated that OVCA4-mKate2 and OVCA8-mKate2 cells were successfully fused with mKate2 protein. Using these cells, we collected mKate2-labeled exosomes for use in the following studies.

### 2.2. Characterization of Exosomes Derived from OVCA4-mKate2 and OVCA8-mKate2 Cells

To investigate the tropism of tumor-derived exosomes in vitro and in vivo, exosomes released by OVCA4-mKate2 and OVCA8-mKate2 cells were isolated by ultracentrifugation and filtration and characterized [3]. Dynamic light scattering (DLS) gives a measurement based on intensity distribution, with the mean size of exosomes derived from OVCA4-mKate2 cells at ~239 ± 51 nm and polydispersity (PDI) 0.343 (Figure 2A). Nanoparticle tracking analysis (NTA) shows the number-weighted distribution of nanoparticles, with a mean size of exosomes derived from OVCA4-mKate2 cells at ~170 ± 48 nm (Figure 2B). Transmission electron microscopy (TEM) of OVCA4-mKate2 exosomes revealed their morphology and ~200 nm size (Figure 2C). Detection of fluorescence of OVCA4-mKate2 exosomes and OVCA4 exosomes at 588 nm shows that OVCA4-mKate2 exosomes have markedly higher fluorescent signals than OVCA4 exosomes, indicating that OVCA4-mKate2 exosomes have abundant mKate2 protein (Figure 2G). Similarly, for OVCA8-mKate2 exosomes, DLS demonstrates that the mean size is ~192 ± 29 nm and PDI is 0.264 (Figure 2D). The number-weighted distribution detected by NTA shows that the mean size is ~141 ± 56 nm (Figure 2E), and TEM shows the morphology and size (Figure 2F). The cup-shaped morphology of exosomes seen in Figure 2F is an artifact caused by dehydration during sample preparation for TEM [40]. Fluorescent values at 588 nm also show that exosomes from OVCA8-mKate2 cells have greater signals than exosomes from OVCA8 cells, indicating a high abundance of mKate2 protein in OVCA8-mKate2 exosomes (Figure 2H). Western blotting of mKate2 protein shows mKate2 bands (~27 kDa) in exosomes derived from OVCA4-mKate2 and OVCA8-mKate2 cells but not in the exosomes from OVCA4 and OVCA8 cells (Figure 2I). This western blot result is consistent with fluorescence detection of mKate2 protein in exosomes derived from OVCA4-mKate2 (Figure 2G) and OVCA8-mKate2 (Figure 2H) cells at 588 nm. Both CD63 and CD9, as exosome protein markers, were detected in all exosome samples (Figure 2J).

The intensity, size, and morphology of exosomes released by OVCA4-mKate2 and OVCA8-mKate2 cells were examined by DLS, NTA, and TEM. We confirmed that these exosomes expressed a more abundant mKate2 protein than exosomes released by non-transfected OVCA4 and OVCA8 cells.

### 2.3. Quantification of Exosomes Released from Attached Cells and Spheroids of OVCA4-mKate2 and OVCA8-mKate2

To quantify the uptake of fluorescent exosomes by cancer cells in vitro or in vivo, a fluorescent calibration curve of known amounts of exosomes was necessary. A standard fluorescent curve for OVCA4-mKate2 exosomes was generated by plotting the fluorescent values of mKate2 protein at 588 nm against the amount of loaded exosomes (Figure 3A). The curve showed a linear relationship between the amount of mKate2 protein and the number of OVCA4-mKate2 exosomes, with a straight line fitted (Y = 3.59 × 10^−8^ × X + 4.34), the R^2^ = 0.99 indicating that 99% of the data fit the linear regression model.

To determine the absolute number of exosomes produced by a single cell, the number of exosomes was compared with the standard fluorescent curve and then divided by the number of cells (Figure 3B,C,E,F). The number of exosomes released per OVCA4-mKate2 attached cell after 1 h of incubation was found to be 51 ± 11, which was greater than those found after 6h (23 ± 09) or 24 h (29 ± 07) incubation (Figure 3B, *n* = 4). In the case of spheroids, exosomes released per OVCA4-mKate2 cell showed a significant increase with time; after 1h, we found 51 ± 25 compared with 73 ± 63 after 6 h and 229 ± 59 after 24 h incubation (Figure 3C, *n* = 6).

A similarly generated fluorescent standard curve for OVCA8-mKate2 exosomes (Figure 3D) indicated a linear regression of amounts of mKate2 protein with numbers of OVCA8-mKate2 exosomes with a straight line fitted (Y = 2.32 × 10^−8^ × X − 0.04). Meanwhile, R^2^ = 0.97 indicated that 97% of the data fit the linear regression model.

Using attached cells to investigate the exosomes released per OVCA8-mKate2 cell, we found 536 ± 85 after 6 h of incubation, which was greater than that produced after 1 h (359 ± 116) or 24 h (392 ± 102) (Figure 3E, *n* = 5–8). For spheroids, exosomes secreted per OVCA8-mKate2 cell were 337 ± 132, 458 ± 66, and 520 ± 95 per cell after 1, 6, and 24 h, respectively (Figure 3F, *n* = 5–6). The more exosomes released per cell in suspended spheroids than attached cells (Figure 3B,C,E,F) may result from more media contacting the cell membrane in the suspended spheroids than in attached cells. The difference in released exosomes between attached cells and spheroids appeared to be more significant with increasing time, i.e., it was highest in the 24 h incubation groups (Figure 3).

The absolute numbers of exosomes released per OVCA4 cell using spheroids (~50–230 exosomes/cell) were significantly higher than those released by attached cells (~20–50 exosomes/cell). This could be because spheroids have enhanced communication between cells, as well as between cells and the extracellular matrix [41,42,43]. However, in the case of OVCA8 cells, the secreted exosome numbers were similar between spheroids and attached cells (~300–500 exosomes/cell). Further studies are needed to address this difference between cell lines.

### 2.4. Uptake of mKate2-Exosomes by OVCA4 and OVCA8 Cells Compared to FTE Cells

It is established that the tropism of tumor-derived exosomes for cancer cells could provide a potential drug-delivery system that specifically targets cancer cells with packaged drugs for enhanced cancer therapy [13]. This was demonstrated using cultured OVCA4 and OVCA8 cells, as well as FTE cells (non-cancer cells), which have been considered as possible origins of high-grade serous ovarian cancers [44,45]. To investigate whether OVCA4-mKate2 exosomes are specifically taken up by OVCA4 cells, both FTE and OVCA4 attached cells were treated with known amounts of OVCA4-mKate2 exosomes. The uptake of OVCA4-mKate2 exosomes by attached OVCA4 cells was ~9.7-fold higher than by attached FTE cells after 1 h treatment (Figure 4A). After 6 h treatment, OVCA4-mKate2 exosomes remained ~3.5-fold greater in attached OVCA4 cells than in attached FTE cells (Figure 4A). When OVCA4 and FTE spheroids were treated for 1, 6, or 24 h, OVCA4-mkate2 exosomes were ~2.2-fold, ~3.9-fold, and ~7.7-fold greater in OVCA4 spheroids than in FTE spheroids, respectively (Figure 4B). With respect to OVCA8-mkate2 exosomes, for attached cells after 1 h and 6 h treatments, exosome levels were ~2.6-fold and ~2.8-fold greater in attached OVCA8 cells than in attached FTE cells, respectively (Figure 4C). For OVCA8 and FTE spheroids, OVCA8-mKate2 exosome levels were also greater in OVCA8 spheroids than in FTE spheroids, with ~8.6-fold, ~3.3-fold, and ~2.4-fold differences, respectively, observed in the 1, 6, or 24 h treatment groups (Figure 4D).

Significant differences in exosome levels were seen in the 24 h incubation groups when comparing spheroids of ovarian cancer cells, either OVCA4 (Figure 4B) or OVCA8 (Figure 4D), with FTE spheroid. However, for the attached ovarian cancer cells, both OVCA4 (Figure 4A) and OVCA8 (Figure 4C), there was no significant difference from FTE cells at 24 h incubation. This may be because exosomes mediate the communication between receipt cells or between receipt cells and extracellular environments, leading to increased homing of exosomes by spheroids [46], although this warrants further investigations. Nevertheless, exosomes derived from OVCA4 and OVCA8 cells were internalized at greater levels by OVCA4 and OVCA8 cancer cells, respectively, than by FTE cells, leading us to speculate that tumor-derived exosomes may represent an effective delivery system to target MDR in cancer therapy [13,47]. In addition, we used OVCA4 cell derived exosomes stained with BODIPY dye to investigate cancer cell-derived tropism within different ovarian cancer cell lines. It is evident that markedly greater fluorescent signals of OVCA4 exosomes were observed in OVCA4 cells compared with A2780 ovarian cancer cells (Figure 4E), further confirming cancer cell exosome-mediated tropism.

### 2.5. Distribution and Quantification of mKate2-Exosomes in OVCA4 and OVCA8 Tumor Mice

We next assessed the distribution of exosomes in vivo in a mouse ovarian tumor model. OVCA4-mKate2 exosomes were injected into OVCA4 tumor mice, and 1, 6, or 24 h later, the mice were euthanized, and brain, heart, lung, spleen, kidney, muscle, tumor, and liver tissues were collected for fluorescent detection of exosomes. The yellow and red colors in tissues in Figure 5A indicate the radiant efficiency values for fluorescence in each tissue. The comparison of fluorescence signals of OVCA4-mKate2 exosomes in different tissues showed that the exosomes were primarily in the brain, liver, and OVCA4 tumor tissues (Figure 5B). Studies of exosome function in the nervous systems show that exosomes involved in physiological and pathological activities in the brain are able to cross the blood-brain barrier [48], consistent with our finding of fluorescent signals from OVCA4-mKate2 exosomes in brain tissue (Figure 5A,B). To quantify the amounts of OVCA4-mKate2 exosomes in OVCA4 tumor, a fluorescent calibration curve of known amounts of OVCA4-mKate2 exosomes was generated by plotting the radiant efficiency values of mKate2 protein at 603 nm against exosomes numbers (Figure 5C), showing a linear relation of amounts of mKate2 protein with numbers of OVCA4-mKate2 exosomes with a straight line fitted (Y = 0.002 × X + 4.9 × 10^6^), with R^2^ = 0.99 indicating that 99% of the data fit the linear regression model. In Figure 5D, it is clear that the red color indicating radiant efficiency of fluorescence became more intense with increasing numbers of OVCA4-mKate2 exosomes. OVCA4-mKate2 exosome levels were determined by comparison with the standard fluorescent curve (Figure 5C) and showed that similar quantities of OVCA4-mKate2 exosomes were found in OVCA4 tumor mice after exosome treatment for 1 (19 × 10^7^ ± 13 × 10^7^), 6 (53 × 10^7^ ± 40 × 10), or 24 h (69 × 10^7^ ± 27 × 10^7^) (Figure 5E, *n* = 3). OVCA4-mKate2 exosome levels in the liver were highest after 24 h treatment (83 × 10^7^ ± 13 × 10^8^), lower in the 6 h (11 × 10^7^ ± 17 × 10^7^) treatment group, and not estimated in the 1 h treatment group. OVCA4-mKate2 exosome levels in the other mouse tissues were below the detection limit.

Similarly, determining the distribution of OVCA8-mKate2 exosomes in the brain, heart, lung, spleen, kidney, muscle, tumor, and liver tissues of OVCA8 tumor-bearing mice showed the exosomes to be primarily in the OVCA8 tumors, with relatively minor signals seen in liver tissues (Figure 5F). The pooled data in Figure 5G indicated that the fluorescent signals of OVCA8-mKate2 exosomes in OVCA8 tumors were markedly higher than in other tissues. To obtain the amounts of OVCA8-mKate2 exosomes in OVCA8 tumors, a fluorescent calibration curve of known amounts of OVCA8-mKate2 exosomes was generated by plotting the radiant efficiency values of mKate2 protein at 603 nm against exosome numbers (Figure 5H), showing a linear relation of amounts of mKate2 protein with numbers of OVCA8-mKate2 exosomes with a straight line fitted (Y = 0.001 × X + 5.0 × 10^6^). Meanwhile, R^2^ = 0.99 indicated that 99% of the data fit the linear regression model. Figure 5I shows that the red color intensity increases with increasing numbers of OVCA8-mKate2 exosomes detected. OVCA8-mKate2 exosome numbers were determined by comparison with the related fluorescent standard curve (Figure 5H), revealing that similar levels of OVCA8-mKate2 exosomes were found in OVCA8 tumor-bearing mice after treatment for 1 (32 × 10^7^ ± 23 × 10^7^), 6 (42 × 10^7^ ± 11 × 10^7^), or 24 h (48 × 10^7^ ± 46 × 10^7^) (Figure 5J, *n* = 3). Compared with OVCA8 tumor tissue, the amounts of OVCA8-mKate2 exosomes in the rest of the tissues were all under the detection limit. In addition to that, we determined the pharmacokinetic profiles of each exosome by measuring its intensity in blood, and the resultant data were plotted in the semilog graph (Figure 5K,L). The slope (*k*) from each graph is 0.05819 for the OVCA8 exosome and 0.05253 for the OVCA4 exosome, respectively. By using the following equation, we can calculate the half-life (*t*_1/2_) of each exosome (*t*_1/2_ = 0.693/*k*), and the half-life of OVCA8 and OVCA4 exosome is 11.91 h and 13.19 h, respectively.

## 3. Conclusions and Discussion

In this study, after the generation of mKate2 protein-transfected ovarian cancer cells (OVCA4 and OVCA8) (Figure 1), and fluorescent exosomes (Figure 2) were isolated and used to treat monolayer ovarian cells, 3D spheroids, and in vivo tumor-bearing mice. We concluded that the quantitative analysis defined the absolute number of exosomes released by a single ovarian cancer cell, as well as the number of exosomes homing to the parental ovarian cancer cells, thus indicating the tropism of cancer cell-derived exosomes. This quantitative information on exosome tropism could provide hints for cancer diagnosis and therapy.

Our data showed that the OVCA8 cell line released and absorbed more exosomes than the OVCA4 cell line (Figure 3C,F, and Figure 4B,D). We assume that this may reflect the properties of different cell lines, such as doubling time; OVCA8 cells reportedly double every 24–32 h [49,50,51,52], whereas for OVCA4 cells, doubling occurs every 30–44 h [49,50,51].

Although cancer cell-derived exosomes have been widely studied in many biological processes, including tumor progression, MDR, and metastasis, quantitative information on the tropism of cancer cell-derived exosomes is limited [1]. However, such quantitative information is crucial for guiding cancer therapy for the following reasons. First, in order to suppress the cancer progression mediated by cancer cell-derived exosomes, the quantification of tumor-derived exosomes will be vital for guiding doses of drugs targeting cancer exosomes in both preclinical and clinical studies [53]. Second, quantifying tumor-derived exosomes in body fluids could be a sensitive diagnostic biomarker and allow cancer diagnosis at early stages [54]. Finally, quantifying tumor-derived exosomes could be a valuable prognostic biomarker in cancer patients undergoing treatment [53].

In Figure 5, our results (OVCA4 in panels A&B; OVCA8 in panels F&G) clearly show that fluorescent exosomes derived from transfected ovarian cancer cells mostly home to their parental cells. Therefore, given these preferential homing properties of tumor-derived exosomes, tumor-derived exosomes could serve as a key player in cancer therapy (Figure 6). This potential utility of exosomes is supported by the work of Park et al. [55].

Ovarian cancer cells transfected with mKate2 protein secrete exosomes to the extracellular matrix or culture media. These fluorescent tumor cell-derived exosomes can home to the parental ovarian cancer cells or to 3D spheroids due to cancer cell exosome tropism. Our results and others also concluded that the tumors could promote their growth through tumor-derived exosome tropism (Figure 6A) [6,7]. Thus, many current studies have started to inhibit tumor exosome release by using exosome inhibitors (e.g., GW4869, calpeptin, manunycin A, Y27632, D-pantethine, imiparamine) (Figure 6B) [56] or co-treatment of cancer drugs with exosome inhibitors [57]. In addition to that, there might be competition between protumor activity from tumor-derived exosome tropism-mediated tumor growth and antitumor activity of drug delivery to the tumor site using drug-encapsulated tumor exosome tropism. We will further investigate this concern in the next study.

Additionally, there might be a potential advantage using the tropism of tumor exosomes, if any therapeutic systems that specifically target tumor exosomes are developed (Figure 6C). Once these systems can target tumor exosomes, then the complexes would have more chances to home to their parental cancer cells due to tropism, which may cause enhanced therapeutic effects on tumors like “suicide bombs”. In order to conduct effective therapy for targeting tumor-derived exosomes and deliver payloads as much as possible to tumor tissue, therapeutic doses applied could be considered based on our quantitative results that 20~70 × 10^7^ exosomes were absorbed by tumor tissue (Figure 5E,J).

The potential other application of cancer cell exosome tropism for cancer therapy has been demonstrated using ovarian cancer cell-derived exosomes as CRISPR/Cas9 plasmid carriers to specifically deliver cargos to ovarian cancer tumors, leading to apoptosis of cancer cells [58]. Moreover, cancer cell-derived exosomes could be used to deliver exogenous cargos (e.g., chemotherapeutic drugs, siRNA, or miRNA) or endogenous cargos (e.g., using genetically modified tumor-tropic exosomes) to tumor cells for cancer treatment [59]. Since cancer exosomes carry proteins, RNA and DNA molecules, and lipids that reflect their parental cancer cells, ovarian cancer exosomes could be important diagnostic indicators of early-stage noninvasive ovarian cancer [60]. Furthermore, since cancer-derived exosomes have similar molecular molecules to their parental cells, they could serve as antigens for generating dendritic cell vaccines for cancer immunotherapy [61]. Finally, by understanding the tropism of cancer cell-derived exosomes, such cancer exosomes could be used as adjuvants for immunotherapeutic vaccines for various cancers [62].

The limitations of the present study are (1) we could not quantify absorbed and released amounts of cancer exosomes by a single cancer cell of tumor tissue from in vivo tumor mouse model; (2) we need to find a more clinically relevant tumor mouse models because, in this study, we only used athymic nude mice to inoculate human ovarian cancer cells. However, the uptake and release of exosomes might be different between immunodeficient nude mice and the C57BL/6 mouse model.

## 4. Materials and Methods

### 4.1. Materials and Antibodies

OVCA4 cells were provided by Dr. Thomas Hamilton (Fox Chase Cancer Center, Philadelphia, PA, USA), and OVCA8 cells were obtained from National Cancer Institute (NCI, MD, USA) [44]. Fallopian tube epithelium (FTE) cells were obtained from the University of Texas MD Anderson Cancer Center [44]. A2780 ovarian cancer cell line (Cat NO. 93112519) was purchased from Millipore Sigma company. RPMI-1640 medium (Millipore Sigma, St. Louis, MO, USA), Medium 199 (Thermo Fisher Scientific, Waltham, MA, USA), MCBD 105 medium (Millipore Sigma, MO, USA), fetal bovine serum (FBS) (CPS serum, FBS-500), penicillin/streptomycin (Thermo Fisher Scientific, Waltham, MA, USA), trypsin-EDTA (0.05%) (Thermo Fisher Scientific, Waltham, MA, USA), epidermal growth factor (EGF) (Corning, human recombinant, Glendale, AZ, USA), Lipofectamine 3000 reagent (Thermo Fisher Scientific, MA, USA), pmKate2-C vector (Evrogen, Moscow, Russia), G418 (VWR Life Science, Radnor, PA, USA), and 10% DMSO (Invitrogen, Waltham, MA, USA) were used in the study. Anti-mKate antibody (~27 kDa, OriGene, 1:1000, Rockville, MD, USA), anti-CD9 antibody (~25 kDa, Abcam, 1:1000, Waltham, MA, USA), anti-CD63 antibody (~60 kDa, Abcam, 1:1000, Waltham, MA, USA), anti-GAPDH (~37 kDa, Abcam, 1:1000, Waltham, MA, USA) and goat-anti mouse IgG-HRP antibody (Santa Cruz Biotechnology, 1:10,000, Dallas, TX, USA) were used for Western blotting. DAPI (Biotium, 1 µg/mL, Fremont, CA, USA) was used to stain nuclei and BODIPY 493/503 (Invitrogen, Waltham, MA, USA) was used to stain OVC4 exosomes for confocal microscopy.

### 4.2. Cell Culture

Both OVCA4 and OVCA8 cells were cultured in RPMI-1640 medium supplemented with FBS and 1% penicillin/streptomycin. The cell culture medium for OVCA4 cells was supplemented with 20% FBS, and that for OVCA8 cells with 10% FBS. FTE cells were cultured in a mixture of Medium 199 and MCBD 105 medium (1:1 ratio), supplemented with 10% FBS and 1% penicillin/streptomycin, as well as 10 ng/mL EGF.

### 4.3. Stable Transfection of Ovarian Cancer Cell Lines with mKate2 Protein

Lipofectamine 3000 reagent and pmKate2-C vector were used for mKate2 protein infusion. Firstly, OVCA4 and OVCA8 cells (0.5 × 10^6^ cells per well) were seeded on 6-well plates for cell culture until 70–90% confluence of cells was observed. Then, Lipofectamine 3000 reagent (3.75 µL) and a mixture of pmKate2-C vector (2.5 µg) and P3000 reagent (2 µL/µg DNA) were separately diluted in 250 µL RPMI-1640 medium each. The two solutions were then mixed and incubated for 15 min at room temperature. A total volume of 500 µL solution was added to each well and cultured for 2 days at 37 °C. Finally, G418 (VWR Life Science, J847) was diluted in cell culture medium to 500 µg/mL and applied to select the transfected cells. After confirmation of mKate2 protein expression (excitation and emission maximums at 588 nm 633 nm) using a confocal microscope, cells with mKate2 protein expression were expanded and cryopreserved at -80℃, then transferred to liquid nitrogen for long time storage.

### 4.4. Confocal Microscope

OVCA4-mKate2 and OVCA8-mKate2 cells (1 × 10^5^ cells/well) as well as spheroids were seeded on 12-well plates (Cellvis, P12-1.5H-N) and cultured at 37 °C for 24 h with 5% CO_2_ present. After fixation with 4% paraformaldehyde for 15 min, cells were washed with PBS solution. Then, DAPI (1 µg/mL) was applied to stain the nuclei of the cells. Confocal images were obtained using a Zeiss LSM 880 confocal microscope with an Axio Observer stand. 20× 0.8 N.A. and 40× 1.2 N.A. water immersion objectives were used.

### 4.5. Western Blotting

Mammalian cell lysis buffer (GoldBio, St Louis, MO, USA) was added to cells and exosomes for 30 min on ice. After centrifugation at 20,000× *g* for 30 min at 4 °C, protein content of the supernatants was determined using Bradford protein assay (Bio-Rad, Hercules, CA, USA). All samples were analyzed by Western blotting as described previously [63,64]. About 30 µg of total proteins were loaded on 4–15% mini-Stain-Free gels (Bio-Rad, Hercules, CA, USA), and, after transfer to PVDF membranes (Millipore Sigma, 0.2 µm, St. Louis, MO, USA), the membranes were probed for mKate2 protein using an anti-FRP antibody. After labeling by goat-anti-mouse IgG-HRP antibody and mouse anti-rabbit IgG-HRP antibody, membranes were exposed to chemiluminescence substrates (Thermo Fisher Scientific, Waltham, MA, USA) and imaged using a ChemiDoc Touch Imaging System (Bio-Rad, Hercules, CA, USA).

### 4.6. Collection and Characterization of Exosomes

OVCA4-mKate2 and OVCA8-mKate2 cells were expanded in culture media at 37 °C with 5% CO_2_, then cell culture media were replaced by serum-free media for exosome collection and incubated for 1, 6, or 24 h. Serum-free media were centrifuged at 2000× *g* for 15 min; then, the supernatant was filtered by 0.2 µm polyethersulfone membrane (Thermo Fisher Scientific, 725-2520) [65]. The filtrate was centrifuged at 3000× *g* for 70 min using Amicon Ultra-15 Centrifugal Filter Units (Millipore, St. Louis, MO, USA). Exosomes were removed and stored at −20 °C for further experiments. The size distribution of exosomes was characterized by dynamic light scattering (DLS). The concentration and size of exosomes were examined by nanoparticle tracking analysis (NTA) (NanoSight NS300 NTA system). The morphology of exosomes was obtained by transmission electron microscopy (TEM).

### 4.7. Up-Take of mKate2-Exosomes by OVCA4, OVCA8, and FTE Attached Cells and Spheroids

OVCA4, OVCA8 and FTE cells (2 × 10^6^) were seeded on surface treated dishes to produce attached cells and expanded on untreated surface dishes to generate spheroids with cell culture media added. Then, cell culture media of attached OVCA4, OVCA8, and FTE cells were replaced by serum-free media, and OVCA4-mKate2 and OVCA8-mKate2 exosomes (1 × 10^9^) were added. For the OVCA4, OVCA8, and FTE spheroids, after centrifugation, spheroids were re-suspended in serum-free media with OVCA4-mKate2, and OVCA8-mKate2 exosomes (1 × 10^9^) added. After incubation for 1, 6, or 24 h, the media of attached OVCA4, OVCA8, and FTE cells were transferred to 15 mL conical tubes for exosome collection; media of OVCA4, OVCA8, and FTE spheroids were separated by centrifugation and stored in 15 mL conical tubes. Both attached cells and spheroids were treated with trypsin for cell counting using 2-Chip (Bulldog Bio, Portsmouth, NH, USA). Cells and purified exosomes were loaded in 96-well plates for fluorescence detection with an excitation at 588 nm and emission at 633 nm using SpectrumMax M3.

### 4.8. Animal Study

All animal procedures were approved by the Institutional Animal Care and Use Committee (IACUC) of the University of Oklahoma Health Sciences Center. Female athymic nude mice (homozygous, 7-week-old, *n* = 24) were anesthetized with isoflurane, and OVCA8 and OVCA4 cells (1 × 10^7^ cells per mouse) were administered by subcutaneous injection over the right flank. When tumors reached a specific size (~1 cm), OVCA4-mKate2 exosomes and OVCA8-mKate2 exosomes (1 × 10^10^ particles per mouse) were given an intravenous injection in the tail vein (treatment groups). Untreated control groups received no exosomes. One, six, and 24 h after treatment, mice were euthanized by CO_2_ inhalation, and the brain, heart, lungs, liver, spleen, kidneys, skeletal muscle, and tumor were excised, snap-frozen in liquid nitrogen, and stored at −80 °C.

### 4.9. In Vivo Fluorescence Imaging

Frozen mouse brain, heart, lungs, liver, spleen, kidneys, skeletal muscle, and tumor tissues were fluorescently imaged at an excitation wavelength of 603 nm and an emission wavelength of 660 nm using the IVIS Spectrum In Vivo Imaging System. Images were analyzed using living image 4.7 software.

### 4.10. Statistics

A two-way ANOVA was used to analyze the fluorescent difference of exosomes derived from OVCA-mKate2 cells and OVCA cells and compare the uptake of OVCA-mKate2 exosomes between OVCA cells and FTE cells. A one-way ANOVA was used to analyze absolute amounts of exosomes released by single OVCA-mKate2 cells after incubation for 1, 6, and 24 h. In the animal study, comparisons of radiant efficiency of tissues, including brain, heart, lung, spleen, kidney, muscle, liver, and tumor tissue, was analyzed using a two-way ANOVA. The absolute amounts of fluorescent exosomes in mice tumor tissues post-injection of exosomes were analyzed using a one-way ANOVA. Data are presented as mean ± SD, and significance was set at *p* < 0.05. Statistical analysis was performed using GraphPad Prism 6.

## Figures and Tables

**Figure 1 ijms-24-08166-f001:**
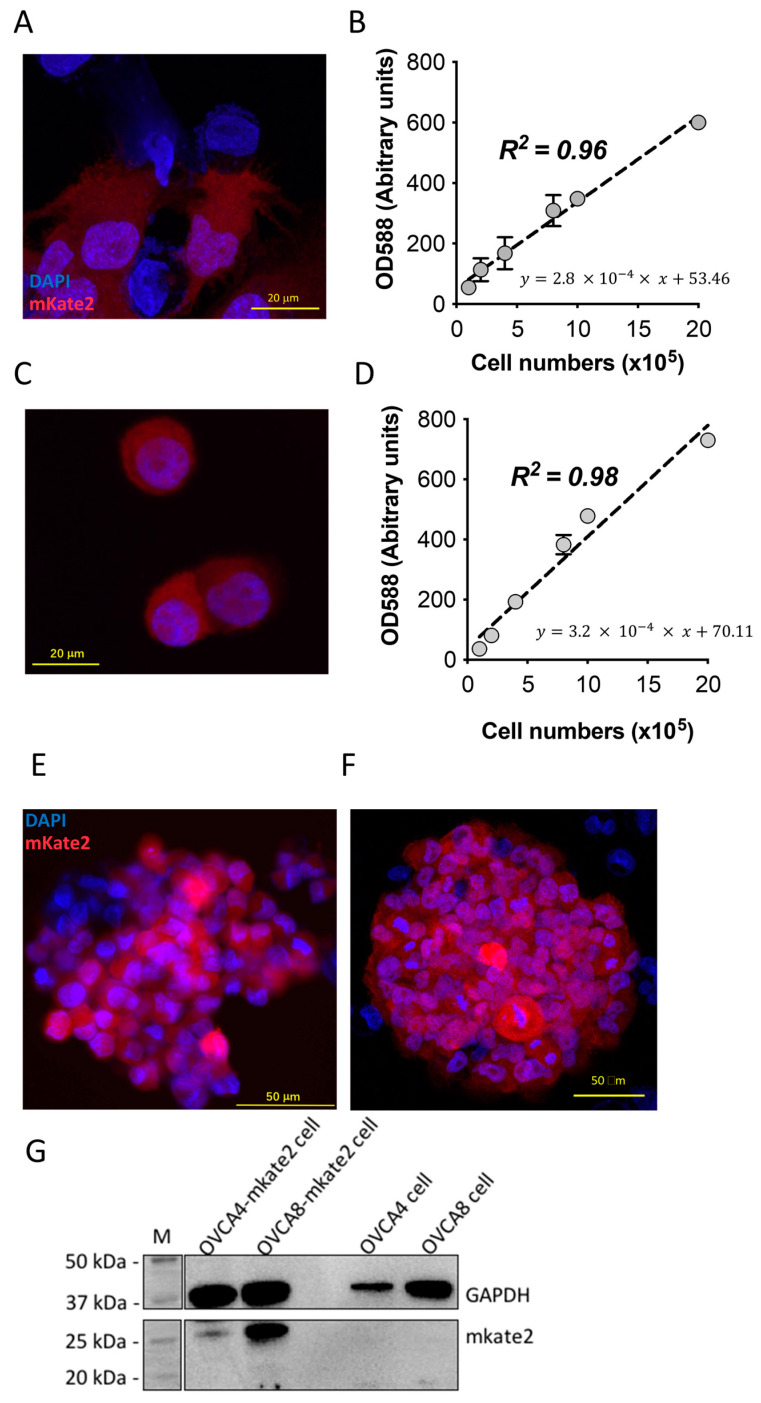
Confirmation of OVCA4-mKate2 and OVCA8-mKate2 cells. Confocal images of OVCA4-mKate2 (**A**) and OVCA8-mKate2 cells (**C**). Fluorescent curves of mKate2 protein in OVCA4-mKate2 cells (**B**) and OVCA8-mKate2 cells (**D**), derived by fluorescent values at 588 nm (*y*-axis) relative to cell numbers (*x*-axis); Values are pooled data and mean ± SD. Confocal images of spheroids of OVCA4-mKate2 (**E**) and OVCA8-mKate2 cells (**F**). The red color indicates mkate2 protein; the blue indicates DAPI staining indicative of nuclei. Scale bars show 20 µm in (**A**,**B**,**E**) and 50 µm in (**F**). Western blot of mKate2 protein (~27 kDa) and GAPDH (~37 kDa) in OVCA4-mKate2, OVCA8-mKate2 cells, OVCA4, and OVCA8 (**G**). GAPDH amount indicates the total protein of cell samples loaded. mKate2 protein, GAPDH, and molecular weight markers (M) as indicated.

**Figure 2 ijms-24-08166-f002:**
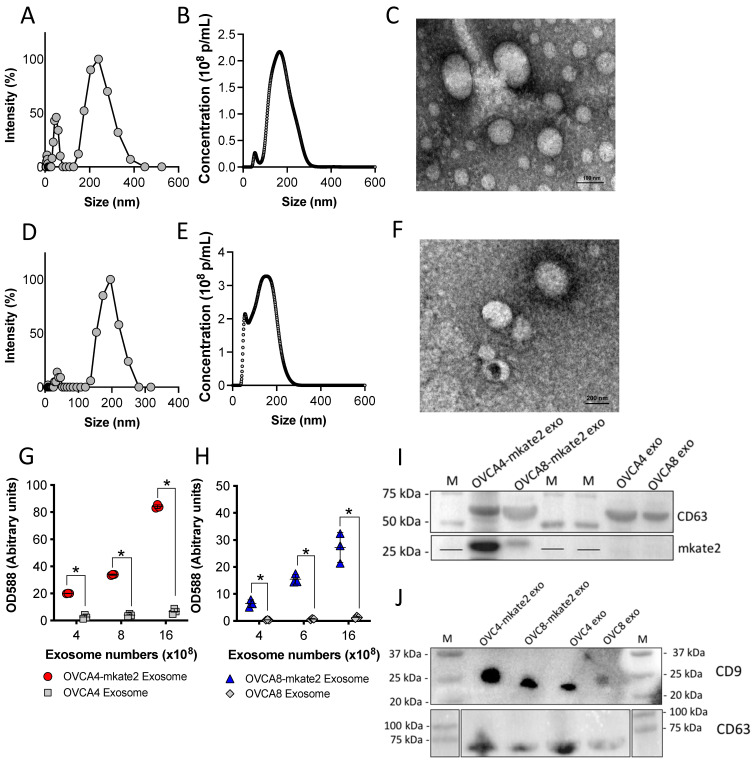
Characterization of exosomes derived from OVCA4-mKate2 and OVCA8-mKate2 cells. Detection of OVCA4-mKate2 exosomes by dynamic light scattering (DLS) (**A**), nanoparticle tracking analysis (NTA) (**B**), and transmission electron microscopy (TEM) (**C**). The TEM image in panel C shows a 100 nm scale bar, taken under 60 kV negative highest voltage and 25,000 × direct magnification. Detection of OVCA8-mKate2 exosomes by DLS (**D**), NTA (**E**), and TEM (**F**). The scale bar in the TEM image in panel G is 200 nm, taken under 60 kV negative highest voltage and 10,000 × direct magnification. Comparison of fluorescence of exosomes derived from OVCA4-mKate2 and OVCA4 cells (**G**). Red round and gray square shapes stand for OVCA4-mKate2 exosome and OVCA4 exosome, respectively (**G**). Comparison of fluorescence of exosomes derived from OVCA8-mkate2 and OVCA8 cells (**H**). Blue triangle and gray diamond shapes indicate the OVCA8-mKate2 exosome and OVCA8 exosome. Western blot of mKate2 protein (~27 kDa) (**I**), CD63 (~60 kDa) (**I**,**J**), and CD9 (~25 kDa) (**J**) in exosomes released from OVCA4-mKate2, OVCA8-mKate2, OVCA4, and OVCA8 cells. mKate2 protein, CD63, and molecular weight markers (M) as indicated. CD63 amount shows the total protein of cell samples loaded. * indicates a significant difference, two-way ANOVA analysis, Holm-Sidak’s multiple post hoc test, *p* < 0.05.

**Figure 3 ijms-24-08166-f003:**
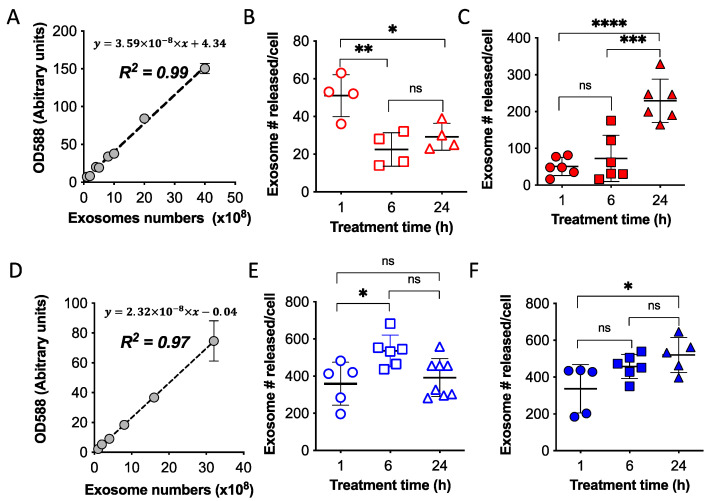
Quantification of exosomes released from transfected attached cells and spheroids of OVCA4 and OVCA8. The fluorescent curve of OVCA4-mKate2 exosomes (**A**) was derived by plotting fluorescent values of mKate2 protein at 588 nm (*y*-axis) against OVCA4-mKate2 exosome numbers (*x*-axis). Absolute amounts of exosomes released by a single OVCA4-mKate2 cell after incubation for 1, 6, and 24 h were detected using attached OVCA4-mKate2 cells (**B**) and OVCA4-mKate2 spheroids (**C**). The fluorescent curve of OVCA8-mKate2 exosomes (**D**) was derived by plotting fluorescent values of mKate2 protein at 588 nm (*y*-axis) against OVCA8-mKate2 exosome numbers (*x*-axis). Absolute amounts of exosomes released by a single OVCA8-mKate2 cell after incubation for 1, 6, and 24 h were detected using attached OVCA8-mKate2 cells (**E**) and OVCA8-mKate2 spheroids (**F**). Values are pooled data and mean ± SD. One-way ANOVA analysis, Holm-Sidak’s multiple post hoc test, *p* > 0.05: no significant difference (ns), *p* < 0.5: *, *p* <0.01: **, *p* < 0.001: ***, and *p* < 0.0001: ****.

**Figure 4 ijms-24-08166-f004:**
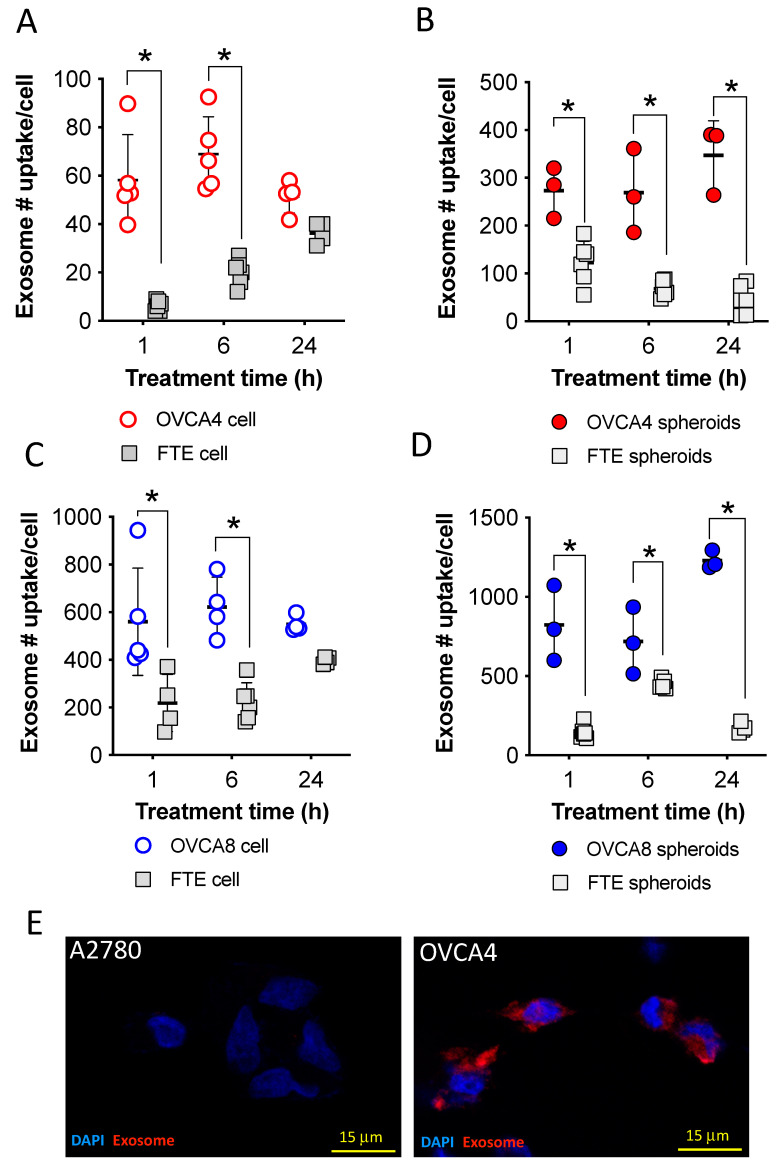
Uptake of mKate2-exosomes by OVCA4 and OVCA8 cells compared to FTE cells. After incubation of OVCA4-mKate2 exosomes for 1, 6, or 24 h, exosome levels were compared between OVCA4 and FTE attached cells (**A**) and between OVCA4 and FTE spheroids (**B**). After incubation of OVCA8-mKate2 exosomes for 1, 6, or 24 h, exosome levels were compared between OVCA8 and FTE attached cells (**C**) and between OVCA8 and FTE spheroids (**D**). Comparison of uptake of OVC4 exosomes stained with BODIPY dye between A2780 ovarian cancer cells and OVC4 cells (**E**). Values are pooled data and mean ± SD. * indicates a significant difference, two-way ANOVA analysis, Holm-Sidak’s multiple post hoc test, *p* < 0.05.

**Figure 5 ijms-24-08166-f005:**
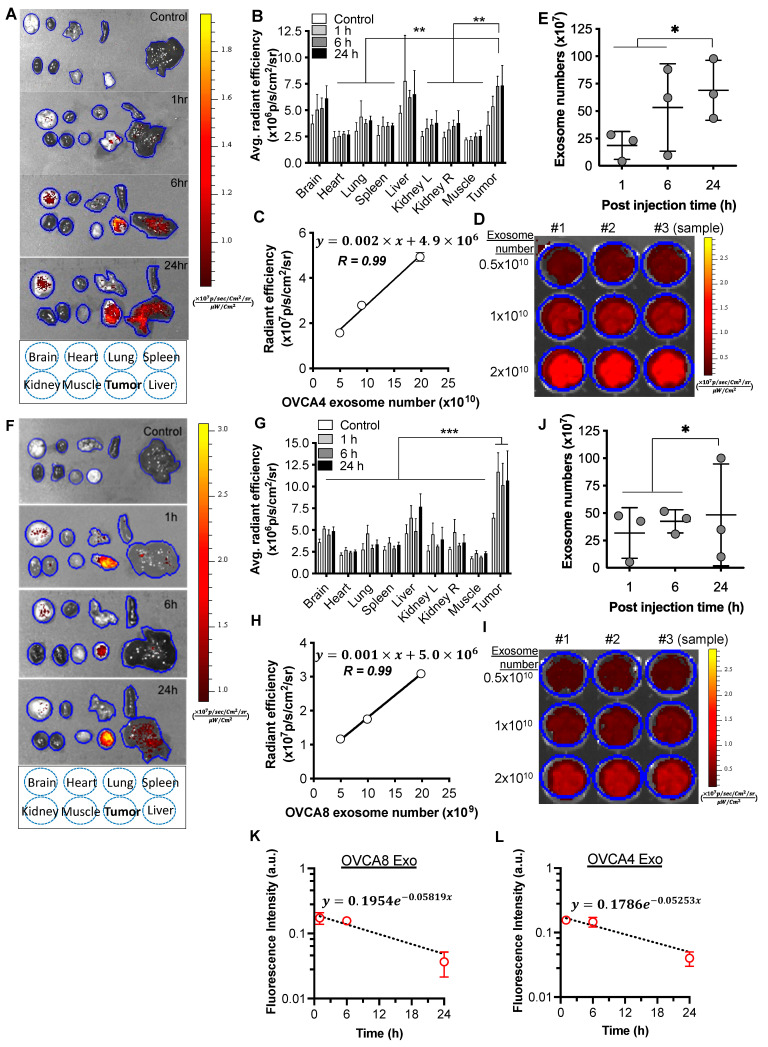
Distribution and quantification of mKate2-exosomes in OVCA4 and OVCA8 tumor-bearing mice. (**A**) Representative fluorescent images of brain, heart, lung, spleen, kidney, muscle, tumor, and liver tissues excised from untreated OVCA4 tumor-bearing mice (Control) and OVCA4 tumor-bearing mice treated with OVCA4-mKate2 exosomes for 1, 6, or 24 h. (Fire bar: fluorescent intensity, yellow is strongest to dark red is minimum intensity). (**B**) Pooled data of average radiant efficiency of different tissues from untreated OVCA4 tumor-bearing mice (white rectangle) and OVCA4 tumor-bearing mice treated by OVCA4-mKate2 exosomes for 1 (light grey rectangle), 6 (dark grey rectangle), or 24 h (black rectangle). (**C**) Fluorescent curves of OVCA4-mKate2 exosomes detected using IVIS imaging system, derived by radiant efficiency at 603 nm (*y*-axis) relative to exosome numbers (*x*-axis). (**D**) Representative image of OVCA4-mKate2 exosomes detected using IVIS imaging System (*n* = 3 samples). (Fire bar: fluorescent intensity, yellow is strongest to dark red is minimum intensity). (**E**) Absolute amounts of the exosomes in OVCA4 tumor tissues after intravenous (IV) injection of OVCA4-mkate2 exosomes for 1, 6, or 24 h. (**F**) Fluorescent detection of OVCA8-mKate2 exosomes in brain, heart, lung, spleen, kidney, muscle, tumor, and liver tissues dissected from untreated OVCA8 tumor-bearing mice (Control) and OVCA8 tumor-bearing mice treated with OVCA8-mKate2 exosomes for 1, 6, or 24 h. (Fire bar: fluorescent intensity, yellow is strongest to dark red is minimum intensity). (**G**) Pooled data of average radiant efficiency of different tissues from untreated OVCA8 tumor-bearing mice (white rectangle) and OVCA8 tumor mice treated by OVCA8-mKate2 exosomes for 1 (light grey rectangle), 6 (dark grey rectangle), or 24 h (black rectangle). (**H**) Fluorescent curves of OVCA8-mKate2 exosomes detected using IVIS imaging System, derived by radiant efficiency at 603 nm (*y*-axis) relative to exosome numbers (*x*-axis). (**I**) Fluorescent image of OVCA8-mKate2 exosomes detected using IVIS imaging system (*n* = 3 samples). (Fire bar: fluorescent intensity, yellow is strongest to dark red is minimum intensity). (**J**) Absolute amounts of OVCA8-mKate2 exosomes in OVCA8 tumor tissues after intravenous (IV) injection of OVCA8-mKate2 exosomes for 1, 6, or 24 h. (**K**,**L**) Determination of the pharmacokinetic profile of exosomes by measuring the exosome intensity in the blood. Values are pooled data and mean ± SD (**B**,**C**,**E**,**G**,**H**,**J**). Two-way ANOVA analysis, Holm-Sidak’s multiple post hoc test, *p* > 0.05: no significant difference, *p* < 0.5: *, *p* < 0.01: **, and *p* < 0.001: ***. (**B**,**G**). One-way ANOVA analysis, Holm-Sidak’s multiple post hoc test, *p* > 0.05: no significant difference, *p* < 0.5: *, *p* < 0.01: **, and *p* < 0.001: ***. (**E**,**J**).

**Figure 6 ijms-24-08166-f006:**
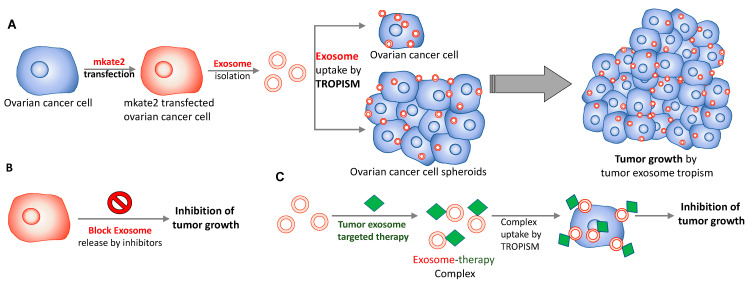
Diagram of cancer therapeutic application using tumor-derived exosome tropism. (**A**) Ovarian cancer cells secrete exosomes to the extracellular matrix or culture media. These fluorescent exosomes can home to their parental ovarian cancer cells or to 3D spheroids through cancer cell exosome tropism, which promotes tumor growth. (**B**) Thus, inhibition of tumor exosome release by the exosome inhibitors could suppress tumor growth prevented by tumor exosome tropism. (**C**) Additionally, potentially, tumor exosome-targeted therapies (green diamond, e.g., exosome-targeted drug delivery systems) could form exosome-therapy complexes, and they are inclined to home to parent cancer cells through tropism and deliver a significant number of therapies to the tumor. Then, finally, it causes to inhibit tumor growth. Thus, a tumor exosome-targeted therapeutic system could be another avenue of drug delivery system as a cancer therapy.

## Data Availability

The data presented in this study is available on request from the corresponding author.

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
