# Peer review of "Understanding of Ovarian Cancer Cell-Derived Exosome Tropism for Future Therapeutic Applications"

_ijms, 2023, doi:10.3390/ijms24098166_

Round 1

Reviewer 1 Report

Reviewer comments

Xiaoyu Ren and the group have worked on the “Understanding of Ovarian Cancer Cell-Derived Exosome Tropism for Future Therapeutic Applications”. According to them exosomes are derived from tumors and have multiple functions in cancer progression, resistance, and metastasis through cancer exosome-derived tropism. They have successfully quantified exosome release and uptake by the parent cancer cells, further proving the tropism of cancer cell-derived exosomes. The finding of this research work implied that the cancer exosome tropism could be useful information for future cancer therapeutic applications. Overall, the paper is interesting and well-planned and accepted after minor corrections as suggested in the comments.

Major comments

1.     The research work is well with good significant results however discussion part is looking weak. Try to incorporate more relevant references in the discussion and compare your data with recently published papers.

Minor comments

1.     In the figure 2 legend, G should come after F however in MS it is mentioned after C. Move the G legend to the right place.

2.     In line 390, give a space between figure 1 and the same in figure 2.

3.     Line 396, write the figures instead of figs.

4.     Line 443 & 476, capitalize the ‘W’ of western.

5.     Quality of figure 3 could be enhanced.

6.     In figure 6, the green symbol is representing which kind of therapeutic molecule? Please mention this in the figure legend. Figures 6B and 6C needed more explanation.

7.     In the discussion part, most of the results are compared with a single reference. Try to compare and discussed more published data to enhance and significance of your results.

8.     Write a conclusion that shows the significance of your results and how your data is support future therapeutic application/s.

9.     In reference 31, write the abbreviated form of the journal name.

Reviewer 2 Report

The authors describe a reuptake strategy of tumor-derived exosomes in ovarian cancer cells, with the future perspective to exploit this mechanism in innovative therapeutic applications. 

Here are my considerations:

1) lines 46-47 reference missed

2) into the introduction I believe that could be interesting to mention the role of tumor exosomes inducing phenotype modification into stromal/mesenchymal cells in order to support tumor growth (author could refer to PMID: 32937811).

3) line 64 the references type "[22-25]" is different to the text 

4)figure 2: why authors did not investigate cd81 and/or cd9 expression on exosomes? They analyzed only cd63, but this could be not so specific. I believe that they should add almost another marker (such as CD81) in order to validate the exosomes. 

5) figure 5A and 5F low quality. Please, increase the pixel resolution. 

6) section 2.6. Therapeutic Applications for Tumor-Derived Exosome Tropism does not describe any results found by the authors but consists of several hypotheses in future applications of tumor-derived exosomes. For this reason, the section needs to be moved into Discussion. 

7) About cell culture protocols, why authors did not use FBS-exosomes depleted? In this way, it is possible to be sure that exosomes released into the cell medium are produced by cells and not already present in FBS supplemented. 

Reviewer 3 Report

This study is interesting with clinical significance. Ovarian Cancer is one of the most common malignant tumor with poor prognosis, which is a clinical problem that needs to be solved. The authors put forward a new point of view to solve this problem via therapeutic applications of exosome. The followings are some comments to the authors.

Comments:

1. I suggest that the mKate2 expression of spheroids of OVCA4-mKate2 and OVCA8-mKate2 cells should be added in Figure 1E

2. What does a,b,b stand for in Figure 3B and 3C? What does ab,b,a and a,a,b stand for in Figure 3E and 3F

3. Is there something wrong with Figure 4E? Please confirm that. I don't understand what the Figure 4E means.

4. How to calculate the surface area of each organ and tumor in Figure 5B and 5G?

5. The fluorescence intensity of tumor is similar to that of liver, and does not show the trend of enhancement. How to explain this phenomenon?

6.The discussion and conclusion can be improved. These kinds of studies have limitations. Hence, the author should have stated the potential limitations and suggested what could be done the next step in this area of research.

Round 2

Reviewer 2 Report

The authors answered all my questions. The article can be accepted for publication.

Reviewer 3 Report

I suggest this manuscript can be accepted in present form.